# The Impact of Shared Assistance between Dermatology and Internal Medicine on Patients with Psoriasis

**DOI:** 10.3390/jcm13082441

**Published:** 2024-04-22

**Authors:** Ana Julia García-Malinis, Juan Blas Pérez-Gilaberte, Tamara Gracia-Cazaña, Maria Pilar González García, Dolores Planas Linares, Yolanda Gilaberte

**Affiliations:** 1Dermatology Unit, San Jorge University Hospital, 22004 Huesca, Spain; lolaplanashu@gmail.com; 2Department of Internal Medicine, Miguel Servet University Hospital, 50009 Zaragoza, Spain; juanblasperezgilaberte@gmail.com; 3Department of Dermatology, Miguel Servet University Hospital, 50009 Zaragoza, Spain; tamgracaz@gmail.com (T.G.-C.); ygilaberte@gmail.com (Y.G.); 4Department of Internal Medicine, San Jorge University Hospital, 22004 Huesca, Spain; pilitagnz@gmail.com

**Keywords:** psoriasis, comorbidities, cardiovascular risk

## Abstract

**Background:** The care of psoriatic patients requires a multidisciplinary approach that addresses not only skin involvement but also cardiovascular risk factors. Coordination between dermatology and internal medicine departments, with a specific focus on treatment and long-term follow-up, can substantially improve the course of a disease and its associated complications. Objective: to evaluate the effects of the holistic management of patients with psoriasis by a multidisciplinary team consisting of dermatology and internal medicine specialists. **Methods:** We conducted an observational, prospective, single-center case–control study between October 2016 and December 2019 in San Jorge University Hospital (Huesca, Spain). Cases included patients undergoing follow-up in the combined dermatology and internal medicine clinic. The control group consisted of an equivalent number of randomly selected, age- and sex-matched patients with moderate-to-severe psoriasis who were seen in the general dermatology department of the same hospital during the same time period. Main outcomes and measures: The primary outcome was the control of psoriatic disease and cardiovascular risk factors such as weight, blood pressure, waist circumference, body mass index (BMI), SCORE index (Systematic Coronary Risk Evaluation), and blood test parameters, as well as diet, physical exercise, and habits such as tobacco and alcohol consumption. To compare data collected over time, data were grouped into three time periods: baseline (t1), intermediate (t2), and final (t3). **Results:** The case group consisted of 27 patients, and the control group consisted of 25 patients. Multivariate analysis was used to evaluate the relationship between the 10-year risk of experiencing a cardiovascular event (SCORE) and the clinical characteristics and analytical variables of patients with psoriasis and controls (*n* = 52). The variables that were significantly associated with a higher 10-year risk of experiencing a cardiovascular event were age (OR, 1.33; CI95% 1.21–1.50; *p* < 0.001); smoking (OR, 5.05, CI95% 1.07–27.37; *p* = 0.047); PASI (OR, 7.98, CI95% 2.32–35.86; *p* = 0.003); BSA (OR, 1.22, CI95% 1.01–1.49; *p* = 0.044); and being a control patient (OR, 3.26; CI95% 0.84–13.56; *p* = 0.029). **Conclusions:** Pharmacological and behavioral interventions carried out as part of the procedure of the multidisciplinary clinic resulted in improvements in the following variables relative to the control group: PASI, BSA, DLQI, PSOLIFE, lipid profile, insulin and HOMA-IR GGT levels, vitamin D levels, and SCORE. These findings indicate the beneficial effect of the multidisciplinary clinic, which reduced the risk of cardiovascular events in psoriatic patients with metabolic comorbidities.

## 1. Introduction

Psoriasis is a chronic, immune-mediated disease characterized by inflammation of the skin and/or joints and is associated with multiple comorbidities [1]. The characteristic inflammation is a consequence of the presence of cytokines (tumor necrosis factor [TNF]-α, interferon [IFN]-γ, and interleukins [IL] 17, 22, 23, and β) that stimulate altered proliferation and differentiation of keratinocytes [2].

Evidence generated in recent years indicates that psoriasis is a cardiovascular risk factor and has a common pathophysiological link to other cardiovascular risk factors, including obesity, diabetes, dyslipidemia, and elevated blood pressure [3]. In 2013, the 67th World Health Assembly categorized psoriasis as a global health problem owing to the morbidity associated with it [4].

Early diagnosis and treatment of psoriatic arthritis is essential to control the disease and prevent progression to joint disability [5]. In recent years, units consisting of dermatologists and rheumatologists have been created to confirm the diagnosis of psoriasis and/or psoriatic arthritis and to agree on treatments in patients with difficult-to-control psoriasis [6]. These units have reported multiple benefits, including greater knowledge of the disease by both health professionals and patients themselves, early diagnosis of psoriasis and psoriatic arthritis, and improved disease management [7,8].

In their systemic review of all-cause and cause-specific mortality risk in psoriasis, Dhana et al. reported that patients with severe psoriasis have an increased risk of all-cause mortality compared with non-psoriatic patients, in part due to an increased cardiovascular mortality risk [9].

The care of psoriatic patients requires a multidisciplinary approach that addresses not only skin involvement but also cardiovascular risk factors and other comorbidities. Coordination between dermatology and internal medicine departments during the management of psoriatic patients, with a specific focus on treatment and long-term follow-up, can substantially improve the course of disease and its associated complications, as well as patient quality of life. In this study, we evaluated the effects of the holistic management of patients with psoriasis by a multidisciplinary team consisting of dermatology and internal medicine specialists.

## 2. Materials and Methods

### 2.1. Study Design

We performed an observational, prospective, single-center case–control study in San Jorge University Hospital (Huesca, Spain). The cases included patients undergoing follow-up in the combined dermatology and internal medicine clinic. The control group consisted of an equivalent number of randomly selected, age- and sex-matched patients with moderate-to-severe psoriasis who were seen in the general dermatology department of the same hospital during the same time period. Control patients underwent no interventions other than those carried out by their dermatologist and primary care physician in routine clinical practice.

Clinical and demographic data were gathered from medical records between October 2016 and December 2019 (Table 1). Patients were informed as to the study objectives, and all provided written informed consent. This study was conducted in accordance with the rules of good clinical practice and approved by the corresponding ethics committee for clinical studies (PI20-479).

### 2.2. Interdiscinplinary Dermatology and Internal Medicine Clinic for Patients with Psoriasis

Patients were simultaneously examined by a dermatologist and an internal medicine specialist in an out-patient clinic located in the dermatology department. Examinations took place on a monthly basis and lasted 3 h, with 3–7 patients evaluated per clinic. Patients were subsequently followed up on every 3 or 6 months depending on their clinical evolution. Patients were referred from the dermatology, rheumatology, and internal medicine departments. Inclusion criteria for admission to the unit were as follows: diagnosis with psoriasis and/or psoriatic arthritis; and 2 or more cardiovascular risk factors (hypertension, hyperlipidemia, diabetes mellitus, smoking and/or obesity) (Table 2). At each follow-up visit, in addition to the control of psoriatic disease, the following were monitored: weight, blood pressure, waist circumference, body mass index (BMI), SCORE index (Systematic Coronary Risk Evaluation), and blood-test parameters (lipid, liver and kidney profile, glycosylated hemoglobin, C-reactive protein [CRP], erythrocyte sedimentation rate [ESR]). Diet, physical exercise, and habits such as tobacco and alcohol consumption were also monitored. The clinic’s staff included a nurse who provided patients with health education related to psoriasis and its associated comorbidities, who surveyed all participating patients about their knowledge of psoriasis and their satisfaction with the multidisciplinary clinic (Figure 1).

### 2.3. Statistical Analysis

Continuous variables were described as the mean and standard deviation, and qualitative variables were described as proportions.

All variables were compared between cases and controls. Quantitative variables were assessed to determine whether they followed a normal distribution, in which case data were expressed as the mean and standard deviation. Non-normally distributed data were analyzed using nonparametric tests and expressed as the median and first and third quartiles. Chi-square tests were used to analyze qualitative variables, and results were expressed as the odds ratio (OR) with a corresponding 95% confidence interval (CI95%) and *p*-value.

To compare data collected over time, data were grouped into 3 time periods: baseline or t1, intermediate or t2 (12–18 months after baseline visit), and final or t3 (12–18 months after intermediate visit). The following comparisons were performed: (i) between-group (cases and controls) for each time period; and (ii) intra-group, within a case or control group, with tests for paired samples for equality and trend (p-trend).

Variables for which statistically significant differences (*p* < 0.05) were observed in the bivariate analysis, as well as possible confounders, were included as independent variables in the multivariate analysis, performed using logistic regression. Results were expressed as the OR and corresponding CI95%, with input criterion *p* < 0.05 and output *p* > 0.1. Statistical analyses were carried out using SPSS software (version 20.0, Armonk, NY, USA: IBM Corp).

## 3. Results

### 3.1. Characteristics of the Sample

We evaluated a total of 52 patients. The case group consisted of 27 patients (22 men [81.5%] and 5 women [18.5%]), with a mean age of 54.19 ± 12.88 years (range, 16–82 years). The control group consisted of 25 patients (18 men [72%] and 7 women [28%]), with a mean age of 56.76 ± 14.42 years (range, 25–78 y). The percentage of smokers was very similar in both groups, while daily alcohol consumption was higher in the case group. More than 50% of patients in the case and control groups had plaque psoriasis (21 [77.8%] and *n* = 15 [60%], respectively; *p* = 0.340), and the mean (SD) duration of psoriasis was 16.48 (10.5) and 16.38 (10.31), respectively (*p* = 0.971). In terms of metabolic comorbidities in the case group, 88.9% (*n* = 24) had dyslipidemia, 55.6% (*n* = 15) were hypertensive, and 25.9% (*n* = 7) were diabetic. Demographic characteristics, the type of psoriasis, and comorbidities are shown in Table 3.

### 3.2. Evolution of Metabolic- and Disease-Related Variables throughout the Study Period

#### 3.2.1. Psoriasis Severity and Quality of Life

As shown in Table 4a, in both cases and controls, PASI and BSA at baseline (t1) were higher than at the end of the follow-up period (t3), with no statistically significant differences between cases and controls. Quality of life, assessed by the DLQI and PSOLIFE questionnaires, was only analyzed in the case group. The mean (SD) DLQI score followed a downward trend, decreasing from 4.04 (5.85) to 2.83 (3.38) (*p* = 0.936), while PSOLIFE scores increased from 69.7 (26.5) to 84.6 (11.8) (*p* = 0.070) in both cases, indicating improved QoL (Table 4b).

#### 3.2.2. Body Mass Index

More than 60% of the patients who attended the dermatology–internal medicine clinic presented some degree of obesity; only 4% of patients had a normal BMI at baseline (t1) (these data were not available for the control group) (Figure 2). The mean BMI in the control group was significantly lower at both follow-up timepoints (t2 and t3) (Table 5) compared with the cases.

#### 3.2.3. Blood Test Variables

Mean (SD) glucose levels were higher in cases than controls, with statistically significant differences observed only at t3 (129 [36.1] mg/dL and 111 [35.2] mg/dL, respectively; *p* = 0.011. Table 6 shows levels of insulin, C-peptide, and HOMA index in patients with psoriasis. Insulin levels (*p* = 0.042) and HOMA index (*p* = 0.138) decreased over time, while C-peptide levels increased (*p* = 0.011) (Table 6a,b).

Mean (SD) cholesterol levels showed no significant differences between groups but exhibited a downward trend over time in patients with psoriasis, decreasing from 201 (43.1) mg/dL at t1 to 182 (37.3) mg/dL at t3 (Table 7). HDL levels were significantly higher in controls than in cases, with significant differences observed at t1 and t3. Mean (SD) LDL levels were similar in controls and cases at t1 (117 (36.3) mg/dL and 115 (43.1) mg/dL, respectively) but significantly lower in cases both at t2 (103 [38.9] mg/dL and 124 (42.2) mg/dL, respectively) and t3 (98.5 (27) mg/dL and 122 (30.6) mg/dL, respectively). Mean (SD) levels of TRG (triglycerides) at t1 were almost 2-fold higher in cases than controls (185 (126) mg/dL and 96 (47.9) mg/dL, respectively; *p* = 0.004) (Table 7), but by t3, had decreased by almost 40 points in cases, from 185 (126) mg/dL to 147 (54.5) mg/dL, and increased slightly in controls, from 96 (47.9) mg/dl to 103 (39.8) mg/dL.

Vitamin D levels were significantly lower at baseline in cases than controls (22.7 ng/mL and 33.5 ng/mL, respectively). At the end of the follow-up period, levels were higher in cases (26.0 ng/mL) than controls (20.9 ng/mL), which were near-deficient in vitamin D (*p* > 0.05; Table 7. GGT levels decreased over time in cases (*p* > 0.05; Table 7).

SCORE (Systematic Coronary Risk Evaluation) value decreased over time in patients with psoriasis cases group. Mean (SD) SCORE at t1 was 2.62 (1.96), indicating a moderate risk of cardiovascular events over 10 years, and it decreased to 1.89 (1.05) at t3. The corresponding decrease in the control group was smaller—from 2.88 (3.30) at t2 to 2.83 (3.34) at t3 (Table 8).

#### 3.2.4. Patient Knowledge of Psoriasis

At the beginning of the study, a survey was conducted to assess the knowledge of both the case and control groups about psoriasis and its associated comorbidities. Approximately 50% of both the control and case groups did not know whether psoriasis could be affected by comorbidities such as blood pressure or cholesterol and vice versa, while 48.1% (n = 13) of cases and 44% (n = 11) of controls did not know that patients with psoriasis could develop psoriatic arthritis in their lifetime. The last two questions, about the benefits of a combined dermatology and internal medicine unit, were only posed to patients with psoriasis who participated in the multidisciplinary clinic. Almost 75% were of the opinion that participation would benefit both their psoriasis and their comorbidities (Table 9a). This same questionnaire was repeated at the final visit (t3) of the patient group. Table 9b shows how knowledge significantly improved for all the questions asked.

#### 3.2.5. Multivariate Analysis

Multivariate analysis was used to evaluate the relationship between the 10-year risk of experiencing a cardiovascular event (SCORE) and the clinical characteristics and analytical variables of patients with psoriasis and controls (n = 52). The variables that were significantly associated with a higher 10-year risk of experiencing a cardiovascular event were age (OR, 1.33; CI95% 1.21–1.50; *p* < 0.001); smoking (OR, 5.05, CI95% 1.07–27.37; *p* = 0.047); psoriasis severity, as measured by PASI (OR, 7.98, CI95% 2.32–35.86; *p* = 0.003); BSA (OR, 1.22, CI95% 1.01–1.49; *p* = 0.044); and being a control patient (OR, 3.26; CI95% 0.84–13.56; *p* = 0.029) (Table 10).

## 4. Discussion

Patients who attended the multidisciplinary dermatology and internal medicine clinic for psoriasis had a long-standing skin disease with approximately three associated comorbidities, most commonly dyslipidemia, obesity, and metabolic syndrome—all of which were associated with a moderate cardiovascular risk. These patients often had unhealthy habits (e.g., smoking and alcohol consumption), and had little knowledge about their disease and associated comorbidities. Pharmacological and behavioral interventions carried out as part of the multidisciplinary clinic resulted in improvements in the following variables relative to the control group: PASI, BSA, DLQI, PSOLIFE, lipid profile, insulin and HOMA-IR GGT levels, vitamin D levels, and SCORE. PASI and BSA were the only parameters for which improvements were observed in the control group. Participation also resulted in a better understanding of psoriasis and its comorbidities among patients with psoriasis.

Psoriasis affects approximately 125 million people worldwide (1–3%) and more than 1 million in Spain (2.69%) [10,11]. It is a systemic inflammatory disease, with synergistic effects with other immune-mediated inflammatory diseases (IMIDs); recognizing the impact of associated comorbidities is therefore essential for comprehensive management.

In 1897, Professor Strauss initially identified the connection between psoriasis and diabetes, sparking research into psoriasis comorbidities [12]. Currently, it is recognized as a systemic condition characterized by chronic inflammation that significantly contributes to its pathology and associated comorbidities [13]. These comorbidities result in increased healthcare costs, diminished quality of life, and a worsened prognosis, prompting significant research focus in recent years [14].

The concept of the “psoriasis march” suggests that systemic inflammation triggered by psoriasis and obesity leads to insulin resistance and dysfunction of the vascular endothelium, subsequently fostering atherosclerosis and the onset of cardiovascular disease (CVD) [15]. Consequently, it is essential to evaluate cardiovascular risk in patients with psoriasis and introduce lifestyle adjustments to regulate blood pressure, glucose levels, and lipid levels. Additionally, maintaining strict therapeutic control is crucial for reducing the systemic inflammation associated with psoriasis. The emergence of biological treatments has revolutionized the prognosis of psoriasis, resulting in enhancements in both skin condition and laboratory parameters [16].

Regarding the severity of psoriasis, it is surprising that both the PASI and BSA scores of both groups at the beginning of the study were mild. This is likely because they were predominantly patients with a long-standing history of the condition who were already receiving systemic or biological treatment under the care of the dermatology department. However, a decrease in both PASI and BSA scores was observed in both groups, although this decrease was more notable in the control group (more than four points on average in terms of PASI) (PASI, *p* = 0.008; BSA, *p* = 0.011). The reason as to why our patients had a poorer response or did not achieve a greater reduction in PASI could be because their BMI was significantly higher than that of the controls. Psoriasis severity has been associated with higher BMI, just as BMI may be a negative prognostic factor for treatment response in psoriasis [17]. Another possibility could be the influence of smoking, which was also higher in the case group. Zhou et al. [18], in a meta-analysis aimed at assessing the associations among smoking and disease risk and treatment efficacy in psoriasis, conclude that smoking negatively influences the benefit of biologic agents; however, they report that more studies are needed to assess the real benefit in the treatment of psoriasis when smoking cessation occurs.

Cardiovascular risk assessment is performed using SCORE, which calculates the 10-year risk of death due to cardiovascular disease from atherosclerotic causes, considering the following factors: age, sex, smoking, total cholesterol levels, and systolic blood pressure. Given the marked geographical variability in CVD in Europe, two SCORE models have been designed, one for high-CVD-risk countries and another for low-CVD-risk countries, the latter of which includes Spain [19].

In this study, both cases and controls had a SCORE of 1–5%, which indicates a moderate 10-year risk. The calculation of overall risk requires a comprehensive patient assessment, which, in addition to SCORE, evaluates risk-modifying factors and data on target organ damage and the presence of CVD. Risk modifying factors include obesity, elevated TRGs, glucose intolerance, and diseases involving increased inflammatory–metabolic stress such as lupus, metabolic syndrome, cancer, HIV, and psoriasis [20]. In our case group, the results of this assessment indicated a high risk, in contrast to the moderate risk estimated based on SCORE alone.

Psoriatic arthritis is a comorbidity of psoriasis of which dermatologists are increasingly aware, in part thanks to the published findings of multidisciplinary units consisting of dermatologists and rheumatologists or internal medicine specialists, which were established first in the USA [21] and subsequently elsewhere [6,7]. Other multidisciplinary care approaches have included, in addition to a rheumatologist and dermatologist, the direct participation of other professionals such as psychologists or nutritionists, with the option of referring to additional specialists, including ophthalmologists, cardiologists, endocrinologists, and digestive specialists [22,23], with the aim of improving patient education as well as disease management [8]. This represents the optimal multidisciplinary approach, to ensure the treatment of psoriasis in a holistic manner, as well as appropriate management of any associated chronic disease.

The impact on the care of psoriatic patients of multidisciplinary units combining dermatologists and internists has not been investigated in depth. However, having a preferential circuit with this medical specialty is one criterion that should be fulfilled for the certification of the quality of psoriasis units [24]. To our knowledge, this study is one of the first to implement and evaluate the benefits of this multidisciplinary approach for patients with psoriasis.

Early screening for comorbidities, coupled with tailored treatment plans, has the potential to enhance the prognosis of individuals with psoriasis. Our multivariate analysis identified age, psoriasis severity (PASI and BSA), smoking, and belonging to the control group as predictors for experiencing a cardiovascular event in 10 years, and implementing a more balanced diet was established as a protective factor. These findings indicate the beneficial effect of the multidisciplinary clinic, which reduced the risk of cardiovascular events in psoriatic patients with metabolic comorbidities.

In chronic diseases such as psoriasis, one of the primary goals of medical treatment is symptom management. Therefore, self-care is essential for controlling symptoms, treatments, psychosocial issues, and quality of life concerns related to the condition. The problem arises when there is insufficient or contradictory knowledge, or when stress or other factors affect treatment adherence. Consequently, providing tailored information and support for each patient’s characteristics in order to enable self-management appears to be a key aspect in psoriatic patient care guidelines [25]. The role of the dermatologist is essential in providing information about the disease and its comorbidities. A management guideline for psoriasis comorbidities states that the role of the dermatologist is essential, not only for early detection but also for informing the patient [3].

The goal of this study was to assess the effects of a more holistic approach to the management of psoriatic patients by simultaneously treating their skin condition and their metabolic comorbidities, as well as increase patient knowledge about their disease both though therapeutic interventions and modifications of diet and habits such as smoking, alcohol intake, and physical exercise. It is well described that extrinsic environmental factors such as alcohol intake, smoking, stress, sleep disturbances, and a sedentary lifestyle, in addition to diet and single nutrient intake, may affect psoriasis clinical presentation and disease severity and course [26,27]. Here is a unmet need to provide patients with accurate science-based information, accessible via online and social media resources, on the influence of extrinsic environmental factors on psoriasis, as well as actively debunk incorrect and unsupported therapeutic recommendations [28]. Research on comorbidities in patients with psoriasis has progressively increased since 2004, with a total of 1803 published articles identified in a bibliographic analysis performed by Huang et al. [29]. These authors recognized screenings for comorbidities, treatment of comorbidities with biologic agents, and multidisciplinary co-management as key future pathways in the psoriasis field.

Why promote holistic management for patients with psoriasis? At the core of our contemporary comprehension of the pathogenesis of psoriasis, there is an interplay among elements of the innate and adaptive immune systems, which is further influenced by diverse external and internal factors, including commensal and pathogenic microorganisms (microbiome) [30,31]. The exposome is composed of two fundamental factors, external factors and internal factors. The main aim of the multidisciplinary units is to treat patients with all the characteristics of their disease and to help them change their habits in order to balance and control their psoriasis. The treatment approach for patients with psoriasis should involve providing education regarding lifestyle modifications and assessing their susceptibility to other comorbidities. There is speculation that reducing circulating cytokine levels may ameliorate the systemic manifestations and complications linked to psoriasis [13]. Understanding the balance between the contributions of the inner and outer psoriasis exposomes would be a step forward in the development of personalized medicine for patients with psoriasis [32].

The limitations of this study include its small sample size and the relatively short follow-up time (<4 years); a minimum follow-up period of 5–10 years is recommended to evaluate the impact of interventions on cardiovascular health in patients with psoriasis [33]. Furthermore, data for certain metabolic variables at baseline were lacking for the control group, and all the patients from the multidisciplinary clinic were included, with no room for randomization, therefore potentially leading to inclusion bias. Another limitation of this study is that the patients in both the case and control groups had low baseline PASI and BSA. This classifies the patients under the mild category or, at most, moderate, but this may be a biased conclusion, as most were on systemic treatment, and there could have been severe cases at baseline.

A key strength of the study is that it is one of very few to evaluate the effects of co-management of patients with psoriasis by a multidisciplinary team.

## 5. Conclusions

In conclusion, we showed that patients with psoriasis with metabolic comorbidities who attended a multidisciplinary unit, combining dermatology and internal medicine, benefitted from improved management of psoriatic lesions, a decrease in cardiovascular risk, and an improvement in most of their comorbidities. Together with overall satisfaction with the multidisciplinary approach, reported by participating patients and professionals, these results support the establishment of similar units to ensure better disease management via a more a holistic approach than is currently available.

## Figures and Tables

**Figure 1 jcm-13-02441-f001:**
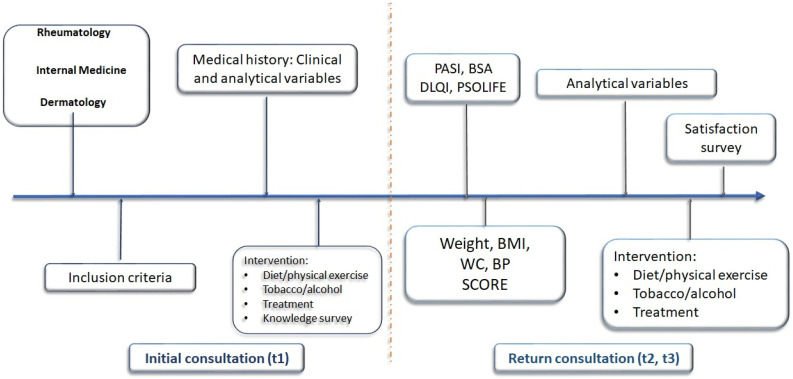
Schematic depicting the multidisciplinary dermatology–internal medicine clinic. Time periods: t1 or baseline, t2 or intermediate (12–18 months after baseline visit), and t3 or final (12–18 months after intermediate visit). (BMI, body mass index; WC, waist circumference; BP, blood pressure, PASI: Psoriasis Area and Severity Index; BSA: Body Surface Area; DLQI: Dermatology Life Quality Index; PSO-life: Psoriasis Quality of Life).

**Figure 2 jcm-13-02441-f002:**
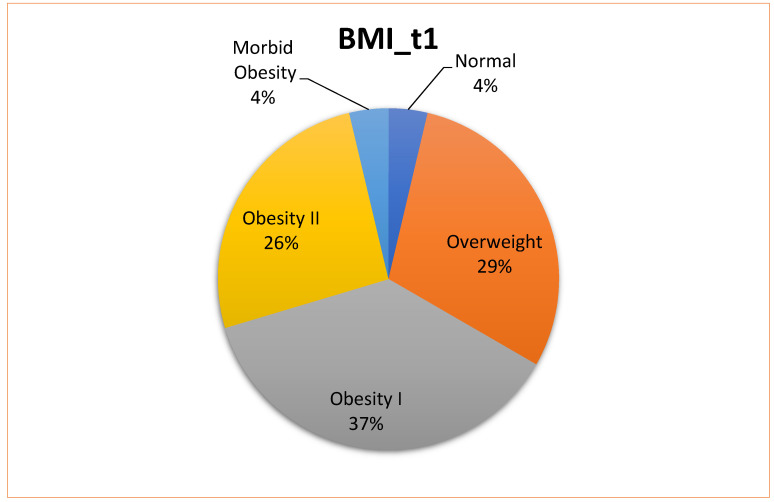
BMI at baseline (t1) in patients with psoriasis who participated in the multidisciplinary dermatology–internal medicine clinic (underweight, <18.5; normal weight, 18.5–24.9; overweight, 25–29.9; class 1 obesity, 30–34.9; class 2 obesity, 35–39.99; class 3 obesity, ≥40).

**Table 1 jcm-13-02441-t001:** Description of clinical and analytical variables collected (PASI: Psoriasis Area and Severity Index; BSA: Body Surface Area; DLQI: Dermatology Life Quality Index: PSO-life: Psoriasis Quality of Life; HbA1c: glycated hemoglobin test; HDL: high-density lipoprotein; LDL: low-density lipoprotein; PTH: parathyroid hormone; ALT: alanine transaminase; AST: aspartate aminotransferase; GGT: gamma-glutamyl transferase; HOMA: homeostatic model assessment).

Clinical Variables
Date of birth, level of education, employment status
First-degree family history of cardiovascular risk factors
Comorbidities: dyslipidemia, hypertension, diabetes mellitus, cardiovascular disease, cancer, hepatopathy, thyroid disease, kidney disease, rheumatologic disease, inflammatory bowel disease, neurological disease, infections, autoimmune disease, psychiatric disease, and other diseases
Habits: smoking or drinking alcohol, type of diet (hypocaloric, Mediterranean, etc.), physical exercise
Psoriasis: type of psoriasis, family history, time since onset, previous treatments and current medication, severity (PASI, BSA), quality of life (DLQI, PSO-life)
Phototype, weight, height, body mass index, waist circumference, blood pressure
Blood test parameters
Glucose, HbA1c, cholesterol, HDL cholesterol, LDL cholesterol, triglycerides, estimated glomerular filtration rate, vitamin D3(25-OH), calcium, intact PTH, uric acid, microalbumin/creatinine quotient, creatinine, ALT, AST, GGT, C-peptide, C-reactive protein CRP, ESR, insulin, HOMA index
SCORE (Systematic Coronary Risk Evaluation): assesses the 10-year risk of dying from a cardiovascular event, as well as high and low cardiovascular risk charts based on gender, age, total cholesterol, systolic blood pressure, and smoking status. Spain is a country with low cardiovascular risk. Scoring: 1%, low risk; 2–4%, moderate risk; 5–9%, high risk; 9–14%, very high risk; >15%, extremely high risk.

**Table 2 jcm-13-02441-t002:** Definition of the cardiovascular risk factors for study inclusion criteria (LDL: low-density lipoprotein; BMI: body mass index).

Arterial hypertension: blood pressure ≥ 140/90 mmHg or antihypertensive treatment
Dyslipidemia: total cholesterol > 200 mg/dL and/or LDL > 130 mg/dL or treatment with a hypolipidemic agent
Diabetes mellitus: glycosylated hemoglobin > 6.5% or blood glucose ≥ 126 mg/mL or treatment with oral antidiabetics or insulin
Obesity: BMI ≥ 30 and/or waist circumference > 80 cm in women and >94 cm in men.
Smoking

**Table 3 jcm-13-02441-t003:** Demographic characteristics, type of psoriasis, and comorbidities (SD: standard deviation) (* statistical significance).

	Cases N (%)	Controls N (%)	*p* Value
Sex			0.630
Male	22 (81.5%)	18 (72%)	
Female	5 (18.5%)	7 (28%)	
Age (mean ±SD)	54.19 ± 12.88	56.76 ± 14.42	0.500
Type of psoriasis			0.340
Plaques	21 (77.8%)	15 (60%)	
Guttate	1 (3.7%)	2 (8%)	
Erythrodermic	0 (0%)	2 (8%)	
Palmoplantar/Nail	5 (18.5%)	6 (24%)	
Time since onset (years)			0.971
(mean ±SD)	16.48 ± 10.5	16.38 ± 10.31	
Smoker			0.338
Yes	9 (33.3%)	10 (40%)	
Ex-smoker	12 (44.4%)	5 (20%)	
No	6 (22.2%)	10 (40%)	
Alcohol			0.016 *
Yes	13 (48.1%)	5 (20%)	
Ex-drinker	0 (0%)	2 (8%)	
Social Drinker	0 (0%)	4 (16%)	
No	14 (51.9%)	14 (56%)	
Number of comorbidities			0.184
Mean ± SD	3.41 ± 1.82	2.64 ± 2.27	
Median	3.00	3.00	
Dyslipidemia			0.001 *
Yes	24 (88.9%)	11 (44%)	
No	3 (11.1%)	14 (56%)	
Time since onset (mean ± SD)	8.44 ± 5.85	10.09 ± 4.5	0.415
Arterial hypertension			0.405
Yes	15 (55.6%)	11 (44%)	
No	12 (44.4%)	14 (56%)	
Time since onset (mean ± SD)	4.66 ± 6.43	11.09 ± 4.48	0.005 *
Diabetes			0.612
Yes	7 (25.9%)	5 (20%)	
No	20 (74.1%)	20 (80%)	
Time since onset (mean ± SD)	3 ± 6.97	12 ± 2.96	0.005 *
Metabolic syndrome			
Yes	17 (62.9%)		
No	10 (37.03%)		
Cardiovascular disease			0.262
Yes (ischemic heart disease, heart failure)	1 (3.7%)	3 (12%)	
No	26 (96.3%)	22 (88%)	
Hepatopathy			0.138
Hepatic steatosis	3 (11.1%)	0 (0%)	
Alcoholic liver cirrhosis	3 (11.1%)	2 (8%)	
HBV	2 (7.4%)	0 (0%)	
No	19 (70.4%)	23 (92%)	
Rheumatologic disease			0.749
Psoriatic arthritis	9 (33.3%)	8 (32%)	
Osteoarthritis	1 (3.7%)	0 (0%)	
Spondyloarthropathy	2 (7.4%)	3 (12%)	
No	15 (55.6%)	14 (56%)	

**Table 4 jcm-13-02441-t004:** (a) Comparison of PASI and BSA between cases and controls over time. (N, number of subjects; PASI: Psoriasis Area and Severity Index; BSA: Body Surface Area). (b) DLQI and PSOLIFE values in patients with psoriasis who participated in the dermatology–internal medicine clinic (DLQI: Dermatology Life Quality Index: PSO-life: Psoriasis Quality of Life) (For DLQI, the lower the score, the better the quality of life; for PSOLIFE, the higher the score, the better the quality of life).

**(a)**							
			**N**	**Mean**	**Standard Deviation**		***p* Value**
PASI_t1	Cases		27	4.43	8.74		0.826
	Controls		24	5.21	6.09		
PASI_t2	Cases		24	3.52	7.21		0.264
	Controls		25	2.22	4.02		
PASI_t3	Cases		23	2.20	3.38		0.256
	Controls		25	1.14	2.56		
BSA_t1	Cases		27	3.99	6.17		0.333
	Controls		24	6.12	7.76		
BSA_t2	Cases		24	3.19	6.91		0.483
	Controls		25	2.82	5.55		
BSA_t3	Cases		23	2.29	3.52		0.345
	Controls		25	1.28	3.26		
	Controls		22	26.9	3.78		
**(b)**							
	**N**	**Mean**	**Standard Deviation**		**Median**	**IQR**	***p* Value**
DLQI_t1	27	4.04	5.85		2.00	[1.00; 4.50]	0.936
DLQI_t2	24	2.67	4.38		1.00	[0.00; 3.00]	
DLQI_t3	24	2.83	3.38		1.50	[1.00; 3.00]	
PSOLIFE_t1	27	69.7	26.5		73.0	[58.5; 91.0]	0.070
PSOLIFE_t2	24	83.0	12.2		84.5	[76.5; 93.2]	
PSOLIFE_t3	24	84.6	11.8		84.0	[78.8; 94.0]	

**Table 5 jcm-13-02441-t005:** Evolution of BMI (body mass index) over time in cases and controls. Underweight, <18.5; normal weight, 18.5–24.9; overweight, 25–29.9; obesity grade I, 30–34.9; obesity grade II, 35–39.99; morbid obesity ≥ 40) (IQR: InterQuartile Range) (* statistical significance).

		N	Mean	Standard Deviation	IQR	*p* Value
BMI_t1 (kg/m^2^)	Cases	27	33.3	5.9	[28.8; 36.3]	-
	Controls		-	-	-	
BMI_t2 (kg/m^2^)	Cases	24	36.1	15.5	[30.3; 35.9]	0.003 *
	Controls	23	27.6	4.23	[25.0; 29.4]	
BMI_t3 (kg/m^2^)	Cases	23	33.9	6.11	[30.0; 36.8]	0.001 *
	Controls	22	26.9	3.78	26.7	

**Table 6 jcm-13-02441-t006:** (a) Glucose and Hb1Ac values in cases and controls over time (normal ranges for glucose: 74–100 mg/dL and Hb1Ac: 0–6.5%) (* statistical significance) (IQR: InterQuartile Range). (b) Insulin, C-peptide, and HOMA-IR values in cases over time (normal ranges: insulin, 3–25 µU/mL; C-peptide, 0.81–3.85 ng/mL [a patient with levels below 0.7 would be considered insulin-dependent]; HOMA-IR, 0–3.8) (IQR: InterQuartile Range) (HOMA: homeostatic model assessment) (* statistical significance).

**(a)**								
		**N**	**Mean**	**Standard Deviation**			**IQR**	***p* Value**
Glucose_t1 (mg/dL)	Cases	27	119	29.8			[98.5; 136]	0.051
	Controls	24	110	32.6			[93.8; 109]	
Glucose_t2 (mg/dL)	Cases	24	118	33.8			[97.2; 124]	0.267
	Controls	25	111	29.5			[92.0; 125]	
Glucose_t3 (mg/dL)	Cases	23	129	36.1			[109; 138]	0.011 *
	Controls	24	111	35.2			[90.0; 110]	
Hb1Ac_t1 (%)	Cases	22	6.25	1.06			[5.60; 6.55]	0.235
	Controls	6	6.88	1.39			[5.80; 7.42]	
Hb1Ac_t2 (%)	Cases	24	6.06	0.85			[5.60; 6.25]	0.762
	Controls	25	6.7	1.55			[6.15; 7.25]	
Hb1Ac_t3 (%)	Cases	23	6.17	0.98			[5.60; 6.40]	0.959
	Controls	24	6.51	1.42			[5.45; 7.50]	
**(b)**								
	**N**	**Mean**	**Standard Deviation**		**Median**	**IQR**		***p* Value**
Insulin_t1 (µU/mL)	21	21.87	15.85		19.4	[15.2; 21.7]		0.042 *
Insulin_t2 (µU/mL)	22	17.76	7.98		14.8	[12.7; 20.6]		
Insulin_t3 (µU/mL)	19	14.24	5.14		15.3	[10.2; 17.1]		
C-Peptide_t1 (ng/mL)	21	2.76	1.31		2.74	[1.90; 3.18]		0.011 *
C-Peptide _t2 (ng/mL)	21	2.95	1.35		3.10	[1.89; 3.49]		
C-Peptide _t2 (ng/mL)	19	3.85	1.53		3.59	[3.08; 4.55]		
HOMA-IR_t1	20	6.14	3.17		5.27	[3.99; 7.78]		0.138
HOMA-IR_t2	20	5.30	3.26		3.81	[3.17; 5.97]		
HOMA-IR_t3	18	4.93	2.70		4.27	[3.34; 6.29]		

**Table 7 jcm-13-02441-t007:** Total values of cholesterol, HDL, LDL, triglycerides, GGT, and vitamin D in cases and controls over time (normal ranges: cholesterol, 150–200 mg/dL; HDL, 40–69 mg/dL; LDL, <70) (optimal; 70–100 (normal), 100–130 (high–normal), >130 (high); TRG, 50–150 mg/dL) (HDL: high-density lipoprotein; LDL: low-density lipoprotein; TRG: triglycerides) (GGT normal range, 0–50 IU/L; minimum recommended vitamin D levels, 20–32 ng/mL) (IQR: InterQuartile Range) (GGT: gamma-glutamyl transferase) (* statistical significance).

		N	Mean	Standard Deviation	IQR	*p* Value
Cholesterol total_t1 (mg/dL)	Cases	27	201	43.1	[170;234]	0.599
	Controls	24	197	36.2	[182; 221]	
Cholesterol total_t2 (mg/dL)	Cases	24	176	35.6	[148; 202]	0.047 *
	Controls	25	200	39.6	[180; 223]	
Cholesterol total_t3 (mg/dL)	Cases	23	182	37.3	[157; 206]	0.437
	Controls	23	191	41.7	[168; 218]	
HDL_t1 (mg/dL)	Cases	27	48.7	13.9	[39.0; 58.5]	0.037 *
	Controls	21	53.7	12.6	[47.0; 56.0]	
HDL_t2 (mg/dL)	Cases	24	46.8	12.7	[38.8; 48.2]	0.062
	Controls	23	54.3	9.94	[48.0; 58.5]	
HDL_t3 (mg/dL)	Cases	19	46.5	11.7	[38.5; 51.0]	0.030 *
	Controls	23	54.3	11.4	[47.0; 61.0]	
LDL_t1 (mg/dL)	Cases	27	117	36.6	[90.2; 145]	0.971
	Controls	21	115	43.1	[85.6; 148]	
LDL_t2 (mg/dL)	Cases	24	103	38.9	[70.7; 124]	0.015 *
	Controls	23	124	42.2	[92.4; 155]	
LDL_t3 (mg/dL)	Cases	24	98.5	27	[75.0; 112]	0.028 *
	Controls	19	122	30.6	[106; 139]	
TRG_t1 (mg/dL)	Cases	27	185	126	[106; 196]	0.004 *
	Controls	21	96	47.9	[72.0; 113]	
TRG_t2 (mg/dL)	Cases	24	176	90.5	[90.0; 218]	0.050
	Controls	23	115	55.6	[77.5; 133]	
TRG_t3 (mg/dL)	Cases	19	147	54.5	[103; 162]	<0.001 *
	Controls	23	103	39.8	[77.5; 122]	
GGT_t1 (UI/L)	Cases	27	80.4	110	[25.0; 59.5]	0.288
	Controls	22	59.9	114	[17.0; 34.5]	
GGT_t2 (UI/L)	Cases	24	72.3	77.0	[28.5; 72.2]	0.153
	Controls	24	56.3	96.8	[16.8; 39.5]	
GGT_t3 (UI/L)	Cases	19	62.5	68.5	[25.5; 63.5]	0.649
	Controls	23	52.9	87.2	[16.0; 42.5]	
Vitamin D_t1 (ng/mL)	Cases	22	22.7	8.96	[17.9; 27.5]	0.008 *
	Controls	7	33.5	29.8	[16.5; 37.5]	
Vitamin D_t2 (ng/mL)	Cases	24	21.2	8.36	[16.0; 25.6]	0.649
	Controls	8	24.8	16.4	[14.0; 28.9]	
Vitamin D_t3 (ng/mL)	Cases	19	26.0	11.2	[18.9; 28.6]	0.215
	Controls	9	20.9	9.54	[13.1; 24.5]	

**Table 8 jcm-13-02441-t008:** SCORE values in cases and controls over time. (Cut-off: <1%, low risk; 1–4%, moderate risk; 5–9%, high risk; 9–14%, very high risk; >15% extremely high risk).

		N	Mean	Standard Deviation	Median	IQR	*p* Value
SCORE_t1	Cases	26	2.62	1.96	2	[1.00; 3.00]	-
	Controls	-	-	-	-	-	
SCORE_t2	Cases	24	2.33	1.49	2	[1.00; 3.25]	0.594
	Controls	24	2.88	3.30	2	[0.75; 3.25]	
SCORE_t3	Cases	19	1.89	1.05	2	[1.00; 2.00]	0.205
	Controls	24	2.83	3.34	2	[0.75; 4.00]	

**Table 9 jcm-13-02441-t009:** (a) Patient knowledge about their psoriasis and comorbidities at the initial visit. (b). Patient knowledge about their psoriasis and comorbidities: comparison between the first and last visit to the dermatology–internal medicine clinic (* statistical significance).

**(a)**			
	**Cases** **N (%)**	**Controls** **N (%)**	***p* Value**
Do you think psoriasis has a genetic predisposition?			0.359
- Yes	9 (33.3%)	13 (52%)	
- No	9 (33.3%)	7 (28%)	
- Do not know	9 (33.3%)	5 (20%)	
Do you think psoriasis only affects the skin?			0.731
- Yes	4 (14.8%)	5 (20%)	
- No	14 (56%)	14 (56%)	
- Do not know	5 (18.5%)	6 (24%)	
Can psoriasis be affected by cholesterol, blood pressure, sugar, or weight?			0.559
- Yes	9 (33.3%)	9 (36%)	
- No	2 (4.4%)	4 (16%)	
- Do not know	16 (59.3%9	12 (48%)	
Can cholesterol, blood pressure, sugar, or weight be affected by psoriasis?			0.875
- Yes	8 (29.6%)	9 (36%)	
- No	4 (14.8%)	3 (12%)	
- Do not know	15 (55.6%)	13 (52%)	
Can all patients with psoriasis have psoriatic arthritis?			0.297
- Yes	4 (14.8%)	8 (32%)	
- No	10 (37%)	6 (24%)	
- Do not know	13 (48.1%)	11 (44%)	
Do you think that the follow-up in this consultation of Dermatology and Internal Medicine will help you control your psoriasis?			
- Yes	20 (74.1%)		
- No	0 (0%)		
- Do not know	7 (25.9%)		
Do you think that the follow-up in this joint consultation of Dermatology and Internal Medicine will help you control your cholesterol, tension …?			
- Yes	20 (74.1%)		
- No	0 (0%)		
- Do not know	7 (25.9%)		
**(b)**			
	**Before** **N (%)**	**After ** **N (%)**	** *p* **
Do you think psoriasis has a genetic predisposition?			0.201
- Yes	9 (33.3%)	14 (58.3%)	
- No	9 (33.3%)	5 (20.8%)	
- Do not know	9 (33.3%)	5 (20.8%)	
Do you think psoriasis only affects the skin?			0.028 *
- Yes	4 (14.8%)	0 (0%)	
- No	18(66.7%)	23 (95.8%)	
- Do not know	5 (18.5%)	1 (4.2%)	
Can psoriasis be affected by cholesterol, blood pressure, sugar, or weight?			0.001 *
- Yes	9 (33.3%)	20 (83.3)	
- No	2 (7.4%)	0 (0%)	
- Do not know	16 (59.3%)	4 (16.7%)	
Can cholesterol, blood pressure, sugar, or weight be affected by psoriasis?			<0.001 *
- Yes	8 (29.6%)	20 (83.3%)	
- No	4 (14.8%)	0 (0%)	
- Do not know	15 (55.6%)	4 (16.7%)	
Can all patients with psoriasis have psoriatic arthritis?			<0.001 *
- Yes	4 (14.8%)	21 (87.5%)	
- No	10 (37%)	0 (0%)	
- Do not know	13 (48.1%)	3 (12.5)	
Do you think that the follow-up in this consultation of Dermatology and Internal Medicine will help you control your psoriasis?			0.007 *
- Yes	20 (74.1%)	24 (100%)	
- No	0 (0%)	0 (0%)	
- Do not know	7 (25.9%)	0 (0%)	
Do you think that the follow-up in this joint consultation of Dermatology and Internal Medicine will help you control your cholesterol, tension …?			0.007 *
- Yes	20 (74.1%)	24 (100%)	
- No	0 (0%)	0 (0%)	
- Do not know	7 (25.9%)	0 (0%)	

**Table 10 jcm-13-02441-t010:** Determinant logistic regression model of the SCORE (PASI: Psoriasis Area and Severity Index; BSA: Body Surface Area).

Coefficients	Estimation	Standard Error	*p*	Odds Ratio (CI95%)
β1 Age	0.290	0.052	<0.001	1.33 (1.21–1.50)
- β2 Smoker				
- No	−0.606	0.879	0.451	0.54 (0.14–4.05)
- Yes	1.621	0.815	0.047	5.05 (1.07–27.37)
β3 PASI	2.077	0.696	0.003	7.98 (2.32–35.86)
β4 BSA	0.197	0.097	0.044	1.22 (1.01–1.49)
β5 Controls–Cases	1.183	0.7015	0.029	3.26 (0.84–13.56)
β6 Diet	−2.669	1.335	0.046	0.06 (0.03–0.83)

## Data Availability

The data presented in this study are openly available in https://zaguan.unizar.es/record/97482/files/texto_completo.pdf (accessed on 2 April 2024) (spanish version).

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
