# Peer review of "The Impact of Shared Assistance between Dermatology and Internal Medicine on Patients with Psoriasis"

_jcm, 2024, doi:10.3390/jcm13082441_

Round 1
Reviewer 1 Report
Comments and Suggestions for Authors
i read with great interest the collaborative effort between dermatology and internal medicine
some suggestions:
1) the study report the holistic appraoch of psoriasis.. more in this direction should be reported as many contributors affect the disease ( characteristics of the patient such as comorbidities - inner and outer exposome) and when a triggering factor occurs create disease flare (https://doi.org/10.3390/vaccines12020178) .so a vicious circle is created with psoriasis promoting cardiovascular disturbances and in turn diseases such as hyperlipedimia etc promotes psoriasis flares and higher disease severity.
2) report what the time periods in the figure text
3) define better the cases and controls in the text.. you report it only in the abstarct
4) line 406 the name of the author is missing
5) in conlusion you report : psoriasis patients with metabolic comorbidities who attended a multidisciplinary unit, combining dermatology and internal medicine specialists, benefitted from improved management of psoriatic lesions, a decrease in cardiovascular risk, and an improvement in most of their comorbidities
well this is anticipated as they were followed by internal medicine clinicians.. the interesting thing that should be discussed is the psoriasis presenations did not present statistical significant improvement?? meaning that this multidisciplinary approach did not improve the disease??
Author Response
Cover letter changes:
Dear Editor and reviewers,
We are resubmitting the reviewed version of our manuscript entitled "Impact of shared assistance between Dermatology and Internal Medicine on patients with psoriasis" (jcm-2973044) to be considered for publication in Journal of Clinical Medicine.
First of all, we want to thank for your time, all your suggestions have helped us to improve our manuscript.
Attending the comments, we have performed some changes highlighted in the manuscript and described in the following paragraphs.
Reviewer 1
Comment 1: the study report the holistic appraoch of psoriasis.. more in this direction should be reported as many contributors affect the disease (characteristics of the patient such as comorbidities - inner and outer exposome) and when a triggering factor occurs create disease flare (https://doi.org/10.3390/vaccines12020178) .so a vicious circle is created with psoriasis promoting cardiovascular disturbances and in turn diseases such as hyperlipedimia etc promotes psoriasis flares and higher disease severity.
Author reply, comment 1: Taking into account the reviewer's suggestion, we have added this paragraph in the discussion:
Why holistic management of the psoriasis patient? At the core of our contemporary comprehension of psoriasis pathogenesis, there is an interplay among elements of the innate and adaptive immune systems, further influenced by diverse external and internal factors, including commensal and pathogenic microorganisms (microbiome)30,31. The exposome is composed of two fundamental factors, external factors and internal factors. The main aim of the multidisciplinary units is to treat patients with all the characteristics of their disease and to help them change their habits in order to balance and control their psoriasis disease. The treatment approach for psoriasis patients should involve providing education regarding lifestyle modifications and assessing their susceptibility to other comorbidities. There is speculation that reducing circulating cytokine levels may ameliorate the systemic manifestations and complications linked to psoriasis13. Understanding the balance between the contributions of the inner and outer psoriasis exposomes would be a step forward in the development of personalized medicine for psoriasis patients32.
Comment 2: report what the time periods in the figure text
Author reply, comment 2: we have added the following sentence at the caption of the figure: Time periods: t1 or baseline, t2 or intermediate (12–18 months after baseline visit) and t3 or final (12–18 months after intermediate visit).
Comment 3: define better the cases and controls in the text.. you report it only in the abstract
Author reply, comment 3: In the material and method section, within the study design, both cases and controls are defined in the following paragraph. I do not know if it is necessary to add any further clarification. We are open to any kind of suggestions in order to improve the quality of the article. Line 77- 83:
“Cases were patients undergoing follow-up in the combined dermatology and internal medicine clinic. The control group consisted of an equivalent number of randomly selected, age- and sex-matched moderate-to-severe psoriasis patients who were seen in the general dermatology department of the same hospital during the same time period. Control patients underwent no interventions other than those carried out by their dermatologist and primary care physician in routine clinical practice.”
Comment 4: line 406 the name of the author is missing
Author reply, comment 4: We have added the author's name: Huang et al. (26).
Comment 5: in conclusion you report: psoriasis patients with metabolic comorbidities who attended a multidisciplinary unit, combining dermatology and internal medicine specialists, benefitted from improved management of psoriatic lesions, a decrease in cardiovascular risk, and an improvement in most of their comorbidities. well this is anticipated as they were followed by internal medicine clinicians.. the interesting thing that should be discussed is the psoriasis presenations did not present statistical significant improvement?? meaning that this multidisciplinary approach did not improve the disease??
Author reply, comment 5: Both the control group and the case group were quite homogeneous in terms of the type of psoriasis, more prevalent in plaques (15, 60% vs. 21, 77.8%, respectively) and its long evolution, more than 16 years on average (SD=10.31 vs. SD=10.5, respectively). Regarding the severity of psoriasis, to us, we were surprised that both PASI and BSA of both groups at baseline were mild. The reason is probably because these were long-standing patients who were already mostly on systemic or biologic treatment in the dermatology department. Nevertheless, a decrease in both PASI and BSA was observed in both groups, although this decrease was more notable in the control group (more than four points on average in terms of PASI) (PASI, p=0.008; BSA, p=0.011), although this decrease was not significant. We have therefore added the following paragraph in the discussion and limitations:
“Regarding the severity of psoriasis, it is surprising that both the PASI and BSA scores of both groups at the beginning of the study were mild. This is likely because they were predominantly patients with a long-standing history of the condition who were already receiving systemic or biological treatment under the care of the Dermatology Service. However, a decrease in both PASI and BSA scores was observed in both groups, although this decrease was more notable in the control group (more than four points on average in terms of PASI) (PASI, p=0.008; BSA, p=0.011). The reason why our patients had a poorer response or did not achieve a greater reduction in PASI could be because their BMI was significantly higher than that of the controls. Psoriasis severity has been associated with higher BMI, just as BMI may be a negative prognostic factor for treatment response in psoriasis 17. Another possibility could be the influence of smoking, which is also higher in cases. Zhou et al 18, in a meta-analysis aimed at assessing the association between smoking and disease risk and treatment efficacy in psoriasis, conclude that smoking negatively influences the benefit of biologic agents; however, they report that more studies are needed to assess the real benefit in the treatment of psoriasis when smoking cessation occurs.”
“Another limitation of the study is that patients in both the case and control groups had low baseline PASI and BSA. This classifies the patients as mild or at most moderate, but this may be a bias as most were on systemic treatment and could have been severe cases at baseline”

Reviewer 2 Report
Comments and Suggestions for Authors
Dear Authors
Thank you very much for your manuscript. I have some comments as follows;
1. There are too many tables. The authors should try to combine relevant data into one table rather than separating them to a,b,c.
2. Discussion is needed to be improved.
2.1 The authors should focus to discuss the benefits of shared assistance between dermatology and internal medicine according to the title and objective of the study. (Line 332-351 described comorbidities of psoriasis which have already well established.) Why is it important to approach patients holisticly?
2.2 In the limited available sources, how authors recommend dermatologists take care psoriasis paitents holisticly.
3. Did the controls receive health education related to psoriasis and associated comorbidites? If health education is not provided, I am not sure whether it is a good clinical practice. Or the authors meaned that "no specific session about psoriasis knowledge" was provided for control.
Author Response
Cover letter changes:
Dear Editor and reviewers,
We are resubmitting the reviewed version of our manuscript entitled "Impact of shared assistance between Dermatology and Internal Medicine on patients with psoriasis" (jcm-2973044) to be considered for publication in Journal of Clinical Medicine.
First of all, we want to thank for your time, all your suggestions have helped us to improve our manuscript.
Attending the comments, we have performed some changes highlighted in the manuscript and described in the following paragraphs.
Reviewer 2
Comment 1: 1. There are too many tables. The authors should try to combine relevant data into one table rather than separating them to a,b,c
Author reply, comment 1: I have merged Table 6c and 6d. Another option is to include Table 9 as supplementary material since they take up a lot of space.
Comment 2.1 : authors should focus to discuss the benefits of shared assistance between dermatology and internal medicine according to the title and objective of the study. (Line 332-351 described comorbidities of psoriasis which have already well established.) Why is it important to approach patients holisticly?
Author reply, comment 2.1: Taking into account the reviewer's suggestion, we have added this paragraph:
Why holistic management of the psoriasis patient? At the core of our contemporary comprehension of psoriasis pathogenesis, there is an interplay among elements of the innate and adaptive immune systems, further influenced by diverse external and internal factors, including commensal and pathogenic microorganisms (microbiome)30,31. The exposome is composed of two fundamental factors, external factors and internal factors. The main aim of the multidisciplinary units is to treat patients with all the characteristics of their disease and to help them change their habits in order to balance and control their psoriasis disease. The treatment approach for psoriasis patients should involve providing education regarding lifestyle modifications and assessing their susceptibility to other comorbidities. There is speculation that reducing circulating cytokine levels may ameliorate the systemic manifestations and complications linked to psoriasis13. Understanding the balance between the contributions of the inner and outer psoriasis exposomes would be a step forward in the development of personalized medicine for psoriasis patients32.
Comment 2.2: In the limited available sources, how authors recommend dermatologists take care psoriasis paitents holisticly.
Author reply, comment 2.2: Nowadays, dermatologists are increasingly aware of the presence of comorbidities in dermatological diseases, not only in psoriasis but also in other diseases such as hidradenitis suppurativa or atopic dermatitis. In hospitals where this type of consultation is not available, one option is to educate the patient about the existence of comorbidities in psoriasis and to refer to the appropriate specialist if we observe that the patient is obese or has an alteration in blood test results. As has been done for years in the early detection of psoriatic arthritis in psoriatic patients.
Comment 3: Did the controls receive health education related to psoriasis and associated comorbidites? If health education is not provided, I am not sure whether it is a good clinical practice. Or the authors meaned that "no specific session about psoriasis knowledge" was provided for control.
Author reply, comment 3: What we were trying to assess with the survey was whether patients with psoriasis were aware that psoriasis is a chronic disease with multiple comorbidities, and we found that both cases and controls were largely unaware of their disease. The cases receive health education about their comorbidities during their visits to the clinic, both from the doctors and from the nurse who assists us in the clinic.
Psoriatic patients who do not meet the criteria for clinic inclusion (i.e., those without comorbidities) are also educated about their condition during their visit and informed about their increased risk for developing comorbidities such as cardiovascular risk factors. This approach enables them to promptly identify any comorbidity and ensures clinical monitoring during follow-up visits. We have therefore added the following paragraph in the discussion:
“In chronic diseases such as psoriasis, one of the primary goals of medical treatment is symptom management. Therefore, self-care is essential for controlling symptoms, treatments, psychosocial issues, and quality of life related to the condition. The problem arises when there is insufficient or contradictory knowledge, or when stress or other factors affect treatment adherence. Consequently, providing tailored information and support to each patient's characteristics to enable self-management appears to be a key aspect in psoriasis patient care guidelines 25. The role of the dermatologist is essential in providing information about the disease and its comorbidities. A management guideline for psoriasis comorbidities states that the role of the dermatologist is essential, not only for early detection but also for informing the patient 3.

Round 2
Reviewer 1 Report
Comments and Suggestions for Authors
The authors did take my suggestions into consideration and the manuscript has been improved. Well done!
Reviewer 2 Report
Comments and Suggestions for Authors
I do not have further comments.
Comments on the Quality of English LanguageThere are some typos.